# Modified Endothelial Activation and Stress Index: A New Predictor for Survival Outcomes in Classical Hodgkin Lymphoma Treated with Doxorubicin-Bleomycin-Vinblastine-Dacarbazine-Based Therapy

**DOI:** 10.3390/diagnostics15020185

**Published:** 2025-01-14

**Authors:** Fazıl Çağrı Hunutlu, Hikmet Öztop, Vildan Gürsoy, Tuba Ersal, Ezel Elgün, Şeyma Yavuz, Selin İldemir Ekizoğlu, Azim Ali Ekizoğlu, Vildan Özkocaman, Fahir Özkalemkaş

**Affiliations:** 1Division of Hematology, Department of Internal Medicine, Faculty of Medicine, Bursa Uludag University, Bursa 16059, Turkey; vildanterzioglu@hotmail.com (V.G.); tubaersal@uludag.edu.tr (T.E.); elgunezel@hotmail.com (E.E.); seymayavuz2011@hotmail.com (Ş.Y.); vildanoz@uludag.edu.tr (V.Ö.); fahir@uludag.edu.tr (F.Ö.); 2Department of Internal Medicine, Faculty of Medicine, Bursa Uludag University, Bursa 16059, Turkey; hikmetoztop@uludag.edu.tr (H.Ö.); drselinildemir@gmail.com (S.İ.E.); aaekizoglu@uludag.edu.tr (A.A.E.)

**Keywords:** Hodgkin Lymphoma, mEASIX, ABVD, survival outcome

## Abstract

**Background:** Although the cure rates of classical Hodgkin Lymphoma (cHL) are as high as 90% using the current treatment protocols, the prognosis is poor for primary refractory patients. Thus, a biomarker that can predict patients with early progression at the time of diagnosis is an unmet clinical need. Endothelial activation and stress index (EASIX) and its variant modified EASIX (mEASIX) is a scoring system currently used for the prediction of prognosis in hematologic malignancies. This study aimed to investigate the prognostic value of the mEASIX score in newly diagnosed cHL patients. **Methods:** Data from 206 patients who underwent positron emission tomography (PET)-guided doxorubicin, bleomycin, vinblastine, and dacarbazine (ABVD) therapy for cHL between January 2007 and November 2023 were retrospectively analyzed. The prognostic value of the mEASIX score was evaluated using the receiver operating characteristic (ROC) analysis, Cox regression analysis, and the Kaplan–Meier method, and then compared with standard risk assessment methods. **Results:** The median age at diagnosis was 33 years, and the rate of patients in the advanced stage was 67%. ROC analysis determined an optimal mEASIX score cut-off of 17.28, categorizing patients into mEASIX^high^ (47%) and mEASIX^low^ (53%) groups. The 5-year progression-free survival (PFS) (60% vs. 84.3%) and overall survival (OS) (79.6% vs. 95.8%) were significantly lower in the mEASIX^high^ group (*p* < 0.001). Additionally, multivariate analysis showed that the independent variables affecting PFS included the nodular sclerosing subtype (HR: 0.4), bone marrow involvement (HR: 2.6), and elevated mEASIX (HR: 3.1). Independent variables, which had an effect on OS included elevated mEASIX (HR:3.8) and higher IPS-3 scores (HR:1.9). Furthermore, a higher mEASIX score (≥17.28) was identified as an independent variable indicating primary refractory disease (OR: 6.5). **Conclusions:** mEASIX is a powerful and easy-to-access marker for the detection of primary refractory disease and prognosis in newly diagnosed cHL cases.

## 1. Introduction

Hodgkin Lymphoma (HL) is a B-cell malignancy accounting for approximately 10% of all lymphoma cases and 5% of lymphoma-related mortalities [1]. Incidence increases in younger adults and those above 55 years of age and has a bimodal distribution [2,3]. Approximately 95% of all HL cases are diagnosed as classical Hodgkin Lymphoma(cHL) and 5% as nodular lymphocyte predominant B-cell lymphoma (NLPBL) [4,5]. Although cure rates are as high as 80–90% with the advancement in HL treatment, approximately 5–10% of cases persist as primary refractory disease, and 10–30% of patients experience early onset or late relapses [6,7,8]. Owing to the standard of care autologous hematopoietic stem cell transplantation, cure rates remain at 30–50% especially in primary refractory disease [9].

Combined chemoradiotherapy (CMT) in the early stages and positron emission tomography (PET)/computed tomography (CT)-guided treatment approaches in the advanced stages are the most commonly used treatment modalities in HL [10,11]. International prognostic score-7 (IPS-7) and IPS-3 have been extensively used to select the treatment protocol and determine the disease risk at treatment onset [12,13]. Nevertheless, IPS-7 and IPS-3 have proven to be inadequate in predicting recurrence/primer refractory disease [13,14,15]. Interim PET/CT (PET2) is superior to standard risk assessment methods, especially in advanced disease [16]; however, recurrence or progression is detected in approximately 18–24% of PET2-negative patients even in PET/CT-guided therapy studies. This indicates that negative PET2 results alone are not sufficient for risk assessment [8,17,18].

Endothelial activation and stress index (EASIX) is used as a scoring tool for endothelial damage and inflammatory response [19]. EASIX is calculated with the formula (creatinine [mg/dL] × Lactate Dehydrogenase (LDH) [U/L]/platelets [×10^9^/L]) [20]. Endothelial dysfunction and damage also impact the glomerular endothelium, contributing to renal failure and increased creatinine levels [21]. During endothelial damage, there is an increase in LDH secretion [22], coagulation cascade activation and platelet counts decrease due to endothelial activation [23]. The initial studies of the EASIX score focused on veno-occlusive disease (VOD) and acute graft-versus-host disease (aGVHD), both of which are severe complications of allogeneic stem cell transplantation characterized by significant endothelial damage. In these studies, a higher EASIX score was correlated with an increased rate of aGVHD and VOD [24,25]. The modified EASIX score (mEASIX) replaces creatinine in the standard EASIX score with C-Reactive Protein (CRP), serving as a marker for inflammation severity and endothelial damage [26]. High EASIX and mEASIX scores are associated with poor prognosis in patients with COVID-19 infection, and these scores have also been recognized as prognostic markers in advanced liver failure [27,28]. Furthermore, an increasing number of studies have suggested that EASIX could be used as a prognostic marker in cases with low-risk myelodysplastic syndrome, multiple myeloma, and diffuse large B-cell lymphoma (DLBCL) [29,30,31,32].

There is a requirement for simple, easy-to-access markers for patients with HL, which would reflect the pathogenesis of the disease to predict cases that may develop early relapse or progression upon standard PET/CT-guided therapies. Furthermore, there are a few studies that have investigated the early detection of primary refractory disease. The present study aimed to investigate the predictive effect of mEASIX score at the time of diagnosis on primary refractory disease and its prognostic significance in cHL patients on doxorubicin-bleomycin-vinblastine-dacarbazine (ABVD)-based therapy.

## 2. Materials and Methods

### 2.1. Patients and Clinical Information

This study included patients aged 18 years and older, who were followed up between January 2007 and November 2023 due to cHL at the Hematology Clinic of Bursa Uludag University Faculty of Medicine. Only the patients who received at least 1 cycle of ABVD-based chemotherapy for cHL were included in the study. Patients with NLPBL, those that tested positive for HIV, those that had concurrent solid organ malignancy, those with HL who received only curative radiotherapy, those left to palliative follow-up, and those whose treatment was started in another center were excluded. The flow-chart of the patients is presented in Figure 1.

Demographics, comorbidities, clinicopathologic features, laboratory parameters, imaging, pathology results, treatment modalities, and treatment responses were retrospectively retrieved from patient files and the hospital information system. All patients were evaluated by PET/CT or CT at the time of diagnosis and staged by the standard Ann Arbor system (Cotswolds-modified) [33]. Moreover, bulky lesions were defined as mediastinal masses exceeding 1/3 of the maximum intrathoracic diameter and nodal or extranodal masses over 7.5 cm. Based on the German Hodgkin Study Group (GHSG) definition, Stage 3, 4, and Stage 2B patients with extranodal involvement or bulky lesions were categorized as advanced stage. Early-stage patients were categorized into risk groups based on the GHSG criteria. The patients’ IPS-7 and IPS-3 scores were calculated in light of the original study [12,13]. For the advanced-stage patients, 0–2 points were considered as low risk, 3–4 as moderate risk, and 5–7 as high risk based on the IPS-7 score; moreover, 0 points were considered as low risk, 1–2 as moderate risk, and 3 as high risk according to the IPS-3 score. The EASIX score was calculated using the following formula: Lactate dehydrogenase (LDH) [U/L] × Creatinine [mg/dL]/platelet count [10^9^/L]. The mEASIX score was calculated using the following formula: LDH [U/L] × CRP [mg/L]/platelet count [10^9^/L] [20,26,27].

PET2 and end-of-treatment PET/CT were used for response assessment, and results were analyzed using the Deauville 5-point scale (Deauville 5PS). A Deauville 5PS score of 1–3 was considered a complete response, 4 as a partial response, and 5 as a no-response/progression. PET2 and end-of-treatment PET/CT evaluations were performed at the same center. All patients included in the analysis received PET/CT-guided ABVD protocol. Throughout the ABVD-based treatment, patients did not receive any additional immunosuppressive therapy. Early-stage patients were evaluated with PET2, and patients with complete PET2 response were treated with a maximum of 2 more cycles of ABVD and with consolidation radiotherapy based on the occurrence of bulky lesions/risk scale at the time of diagnosis. Furthermore, advanced PET2-negative patients underwent 4 cycles of doxorubicin- vinblastine-dacarbazine (AVD). Patients with PET2 partial response (Deauville 4) received 2–4 cycles of escalated-BEACOPP (bleomycin, etoposide, doxorubicin, cyclophosphamide, vincristine, procarbazine, and prednisone) protocol depending on whether they were in an early or advanced stage. Patients who progressed during treatment or relapsed within 3 months despite complete response at the end of treatment were deemed to have primary refractory disease [34].

### 2.2. Statistical Analysis

Overall survival (OS) was defined as the time from diagnosis until the last follow-up or mortality due to any cause. Progression-free survival (PFS) was calculated from the time of diagnosis to the last follow-up, relapse, disease progression, or mortality due to any cause, whichever occurred first. Descriptive statistics were presented as counts and percentages for categorical variables. Furthermore, for continuous variables, means, and standard deviations were reported when data distribution was normal, otherwise, medians and minimum–maximum values were used. Student’s *t*-test or Mann–Whitney U test was used to compare continuous variables between the two groups, depending on whether the normal distribution hypothesis was met or not. Categorical variables were compared using the Chi-squared test. The optimal cut-off point for the mEASIX score was determined using receiver operating characteristic (ROC) curve analysis to predict disease progression or mortality. Survival curves were designed based on the Kaplan–Meier method and compared using the log-rank test. The Cox proportional hazard regression model was used for univariate and multivariate analyses for PFS and OS. Cox proportional hazard regression model with the Backward LR method was used for multivariate analysis, including factors with a *p*-value of less than 0.2 in univariate analysis. A *p*-value of less than 0.05 was considered statistically significant for multivariate analysis. Furthermore, binary logistic regression was used to analyze variables, which predicted primary refractory disease. Binary logistic regression with the Backward LR method was used for multivariate analysis, including factors with a *p*-value of less than 0.2 in univariate analysis. A *p*-value of less than 0.05 was considered statistically significant for multivariate analysis. The Statistical Package for the Social Sciences (SPSS) Version 29.0 (IBM Corp, Armonk, NY, USA) was used for all analyses.

## 3. Results

### 3.1. General Patients Characteristics

Demographic data and clinicopathologic features of 206 patients included in the study are presented in Table 1. The median age at diagnosis was 33 years, and 52.9% of the patients were men. The most common pathologic subtype was nodular sclerosis (60.7%); moreover, 54.9% of patients had B symptoms, and 10.7% of the study group had bulky lesions at the time of diagnosis. The rate of advanced-stage patients was 67%. The majority of advanced-stage patients (93.7%) were at low risk according to IPS-7, whereas the same rate was 34.1% based on IPS-3. Most patients were at moderate risk according to the IPS-3 score. The rate of patients treated with a chemoradiotherapy combination was 22.3% with a complete response rate of 82% at the end of treatment. The primary refractory disease rate was 11.2% and recurrence was observed in 27 patients during follow-up.

ROC curve analysis aimed to find the cut-off value of the mEASIX score to predict mortality, as depicted in Figure 2. Analysis results revealed that the cut-off value was 17.28, and the power of this value in predicting PFS, OS, and the presence of primary refractory disease is shown in Table 2. Patients with an mEASIX of <17.28 were categorized as mEASIX^low^, and those with an mEASIX of ≥17.28 were categorized as mEASIX^high^.

Patient characteristics by the mEASIX score are presented in Table 3. There was no significant difference between the two groups by age, sex, and pathologic subtype distribution. The occurrence of B symptoms and bone marrow involvement at the time of diagnosis were significantly higher in the mEASIX^high^ group (*p* < 0.001, *p* = 0.009). Moreover, the rate of advanced-stage disease was 91.8% in the mEASIX^high^ group and 45% in the mEASIX^low^ group; this difference was statistically significant (*p* < 0.001). In the mEASIX^low^ group, all the patients were in the low-risk based on the IPS-7 score, whereas this rate was approximately 90% in the mEASIX^high^ group. There was no significant intergroup difference by IPS-3 risk score. The LDH, erythrocyte sedimentation rate and CRP levels correlated with disease activity were significantly higher in the mEASIX^high^ group (*p* < 0.001), whereas lymphocyte count, serum albumin, and hemoglobin levels were significantly lower (*p* < 0.001). The presence of primary refractory disease (19.6% vs. 3.7%), recurrence (17.6% vs. 9.2%), and incomplete response rate (28.8% vs. 8.3%) were significantly higher in the mEASIX^high^ group (*p* < 0.001, *p* = 0.021, *p* = 0.001).

### 3.2. Univariate and Multivariate Analyses for Treatment Response and Survival Outcomes

The results of logistic regression analysis to predict primary refractory disease at the start of treatment are given in Table 4. Multivariate analysis indicated that bone marrow involvement at the time of diagnosis and elevated mEASIX (≥17.28) were the independent variables, which predicted primary refractory disease (*p* = 0.010, *p* = 0.004).

Cox proportional regression analysis results, which reviewed the factors with an effect on PFS and OS, are presented in Table 5 and Table 6. Multivariate analysis results revealed that the nodular sclerosing subtype was associated with decreased risk of progression, and bone marrow involvement and higher mEASIX score (≥17.28 vs. <17.28) increased the risk of progression by approximately three-fold (HR:2.6, HR:3.1, *p* = 0.008, *p* < 0.001). Similarly, a higher mEASIX score increased the risk of all-cause mortality by approximately four-fold (HR:3.8, *p* = 0.004), whereas a higher IPS-3 score increased the risk of mortality by approximately two-fold (HR:1.94, *p* = 0.038). Age at diagnosis, stage, hemogram parameters at diagnosis, biochemical markers, and IPS-7 score were not significant for PFS and OS, as revealed by multivariate analysis.

### 3.3. Survival Outcomes

PFS analysis based on the mEASIX score is illustrated in Figure 3. The median follow-up period of the study group was 57 months, and the 1- and 5-year PFS was significantly lower in the mEASIX^high^ group compared to the mEASIX^low^ group (82.3%, 60% vs. 98.1%, 84.3%, respectively, *p* < 0.001). Upon analysis by stage, in early-stage, the 1- and 5-year PFS rates were 87.5% and 37.5% in the mEASIX^high^ group, and the same were 98.2% and 92.3% in the mEASIX^low^ group, respectively, this difference was statistically significant (*p* = 0.004). Similarly, 1- and 5-year PFS were significantly lower in the mEASIX^high^ group in advanced-stage patients (81.8%, 60.7% vs. 98%, 73%, *p* = 0.039).

OS analysis based on the mEASIX score is depicted in Figure 4. The 5-year OS was significantly lower in the mEASIX^high^ group compared to the mEASIX^low^ group (79.6% vs. 95.8%, *p* < 0.001). Furthermore, the 5-year OS was 95.3% in the mEASIX^low^ group, and this rate was significantly lower at 78.2% in the mEASIX^high^ group in advanced-stage patients (*p* = 0.014). In early stage patients, the 5-year OS rates were similar in both groups (100% vs. 96.2%, *p* = 0.578). The relatively small number of patients in the mEASIX^high^ group (8 vs. 60) and the higher rate of cure in early stage patients with subsequent treatments might have accounted for the fact that there was no intergroup difference between by OS.

The effect of mEASIX and IPS-3 scores on PFS is illustrated in Figure 5. Five-year PFS was found to be significantly lower in the mEASIX^high^ group than in the mEASIX^low^ group regardless of the IPS-3 score (60.2%, 60.7% vs. 82.3%, 86.3%, *p* < 0.001). The analysis of OS based on the mEASIX and IPS-3 scores is depicted in Figure 6. The mEASIX^high^ IPS-3^high^ group had the shortest 5-year OS, whereas the mEASIX^low^ group had a significantly longer 5-year OS of over 90% regardless of the IPS-3 score (*p* < 0.001). Although the 5-year OS was lower in the mEASIX^high^ IPS-3^high^ group than in the mEASIX^high^ IPS-3^low^ group, this difference did not reach statistical significance (72.3% vs. 93.4%, *p* = 0.123).

## 4. Discussion

In the era of novel therapies, it is crucial to identify at the time of diagnosis the group of patients who will respond inadequately to standard protocols [35,36,37]. Neither IPS-7 nor IPS-3, which are often used for risk assessment, are adequate to identify patients at very low and very high risk [13,14,15,38]. The use of inflammatory markers in risk assessment improves prognosis, particularly in Hodgkin Lymphoma, where the inflammatory microenvironment constitutes the majority of the tumor tissue [39,40,41,42]. To the best of our knowledge, this is the first study to evaluate the prognostic significance of the mEASIX score, which is an indicator of inflammation and endothelial damage, in patients with cHL. A higher mEASIX score at diagnosis (≥17.28) was associated with lower PFS and OS as well as a higher rate of primary refractory disease. mEASIX score is an easy-to-calculate, practical, and disease biology-related marker that can be used for risk assessment at diagnosis in early and advanced cHL cases.

PET2 examination provides crucial information with regard to prognosis and precedes the standard scoring systems used in risk assessment at the time of diagnosis [43,44]. Nevertheless, PFS rates are approximately 80%, and a significant rate of patients can be at risk of relapse or progression even in PET2-negative cases [43,45]. The prognosis is worse compared to other patients, and aggressive treatment protocols should be applied in the early stages, especially in primary refractory patients. This group of patients typically do not respond to standard protocols and are referred to autologous stem cell transplantation with salvage therapies. Few studies in the literature investigate this patient group’s risk factors [46]. The ECLIPSE study on 478 patients with primary treatment failure reported that 5-year PFS rate was 50%. Moreover, subgroup analyses revealed that PFS was lower, especially in primary refractory patients compared to other groups [47]. In the present study, a higher mEASIX score at the time of diagnosis was an independent predictor of primary refractory disease. Inclusion of the mEASIX score in the initial risk assessment can help with better identification of high-risk patients who may progress earlier.

Currently, inflammatory scoring is used for risk assessment in most malignancies [48,49]. HL is at the top of the list of malignancies, where the inflammatory microenvironment and the cytokines secreted therefrom are closely associated with prognosis [42,50]. As regards the pathogenesis of HL, the structure consisting of inflammatory cells, which form the tumor microenvironment (TME) and surrounding endothelial cells, tumor-associated macrophages, and mast cells play an important role [51,52]. Alternatively, activated macrophages (M2) stimulate angiogenesis and suppress immunity around tumor cells by expressing CD163 on their surface [53,54]. Similarly, it was reported that mast cells increased angiogenesis in patients with HL. Increased M2 macrophage levels and mast cell mass in tumor tissue were associated with lower treatment response rates and OS [51,55,56,57]. Moreover, previous studies reported that the levels of vascular endothelial growth factor-A (VEGF-A) and circulating endothelial progenitor cells in peripheral blood were significantly higher in newly diagnosed and relapsed HL cases compared to healthy controls, which is another indicator of increased angiogenesis in pathogenesis [58]. The inflammatory biology of HL along with increased endothelial activation and angiogenesis allows the prognostic use of markers of inflammation and endothelial damage [12,59,60,61]. Numerous studies have used inflammatory markers in predicting the prognosis of patients with HL and have reported that high inflammation scores were associated with poor prognosis in early and advanced-stage patients [60,62,63]. A study by Paydas et al., reported that the IPS-4 score, which was a product of inflammatory and nutritional markers integrated into the IPS-3 score, was an independent variable for PFS and OS [41]. In the present study, inflammatory markers were higher in the group with high mEASIX scores, and the prognosis in this group was worse in early and advanced-stage patients.

Kaplan–Meier analyses for PFS and OS of the IPS-4 score developed by Paydas et al. were found to be correlated with the IPS-3 score [41]. Survival analyses of the IPSPLR score by Tao et al. showed a correlation with IPS-3, but it proved more effective in identifying very high-risk and low-risk patients [64]. In our study, mEASIX serves as an independent predictor for both PFS and OS. When assessed alongside IPS-3, it appears to be an effective and easily accessible score for identifying patients at very high risk and very low risk.

Although the first studies of the EASIX score and its variants were regarding cellular therapies, including allogeneic stem cell transplantation and CAR-T cells, it was suggested that they could be used as a prognostic marker in hematologic malignancies with increasing frequency [26,29,30,31,65]. Two studies on patients with DLBCL reported that higher EASIX scores were associated with aggressive disease biology and lower PFS and OS rates [31,32]. In the present study, the original EASIX score had no prognostic significance because renal involvement is rare in patients with HL [66] and the disease has an inflammatory biology [42], whereas a higher mEASIX score was associated with lower treatment response rates and a shorter OS.

The limitations of this study included the fact that it was designed as a retrospective, single-center research, and the mEASIX score was evaluated only at the time of diagnosis without any dynamic follow-up data. Moreover, current prognostic markers, including metabolic tumor volume and circulating tumor DNA could not be evaluated due to insufficient data.

## 5. Conclusions

The mEASIX score at the time of diagnosis is an easy-to-access, practical scoring system that can be used to predict primary refractory disease and prognosis in patients with cHL. High-risk patient groups can be identified more accurately and sooner if the mEASIX score is included in the initial risk assessment. Prospective randomized controlled studies on the mEASIX score with a larger number of patients can help further validate the results of the present study.

## Figures and Tables

**Figure 1 diagnostics-15-00185-f001:**
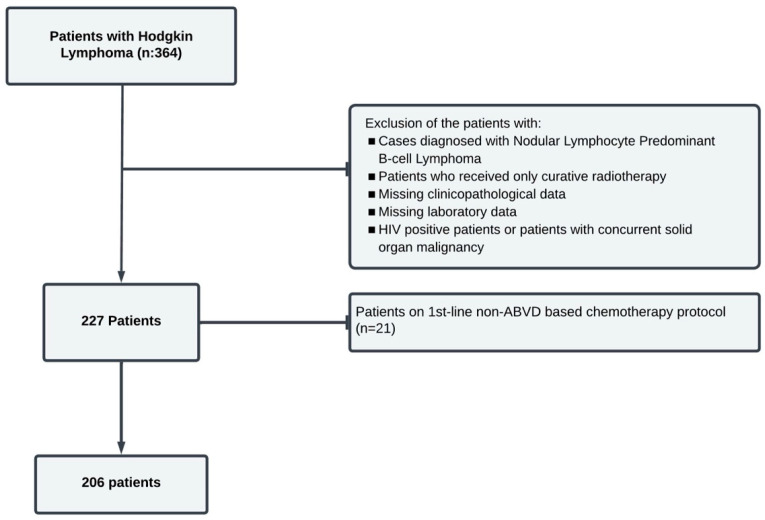
Flow-chart of the study.

**Figure 2 diagnostics-15-00185-f002:**
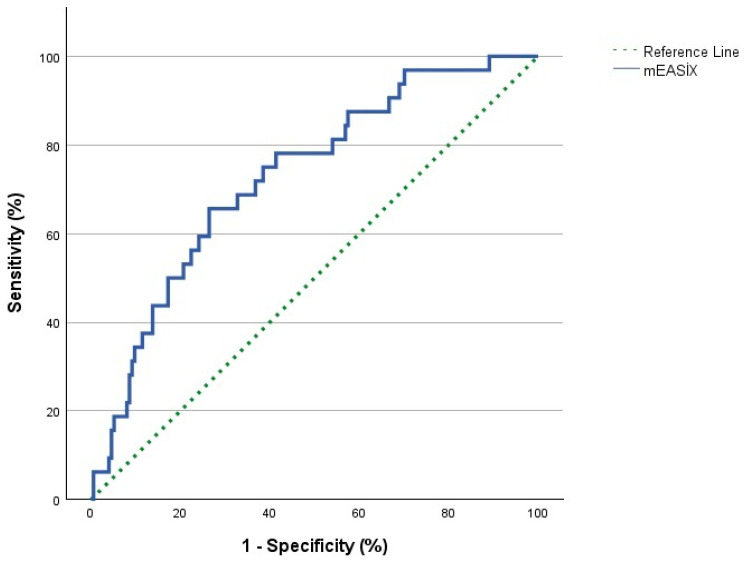
ROC Curve Analysis of mEASIX score.

**Figure 3 diagnostics-15-00185-f003:**
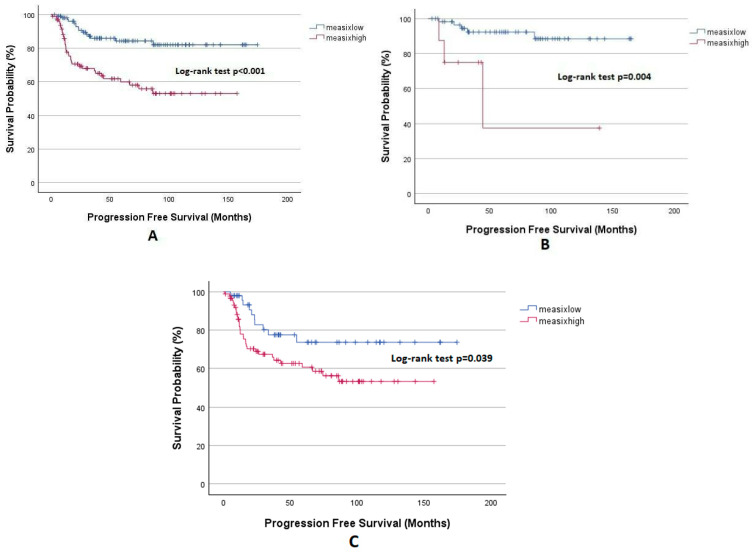
Kaplan–Meier analysis for PFS, (**A**) all patients, (**B**) early stage, and (**C**) advanced stage.

**Figure 4 diagnostics-15-00185-f004:**
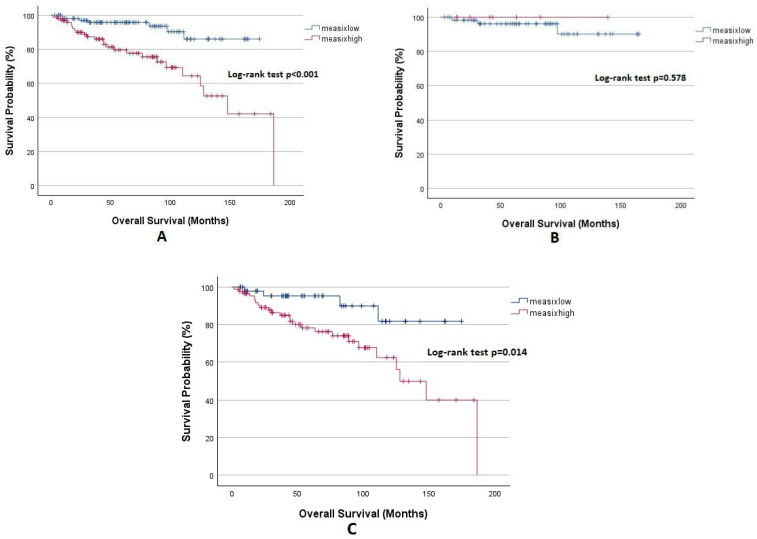
Kaplan–Meier analysis for OS, (**A**) all patients, (**B**) early stage, and (**C**) advanced stage.

**Figure 5 diagnostics-15-00185-f005:**
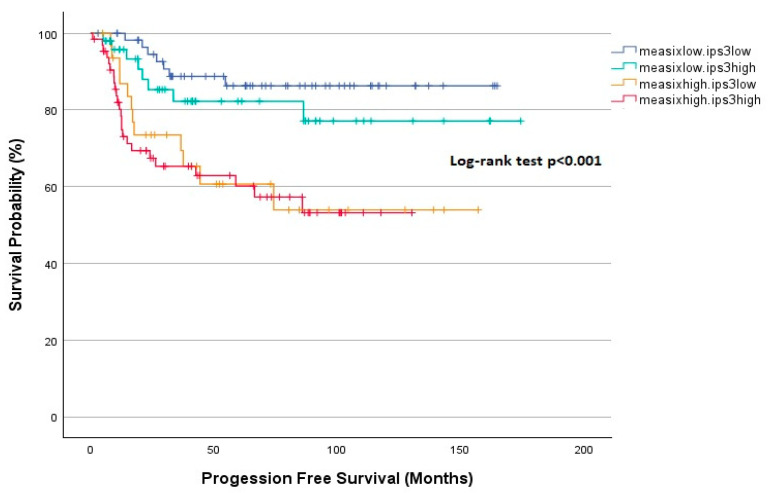
Kaplan–Meier analysis for PFS according to mEASIX and IPS-3.

**Figure 6 diagnostics-15-00185-f006:**
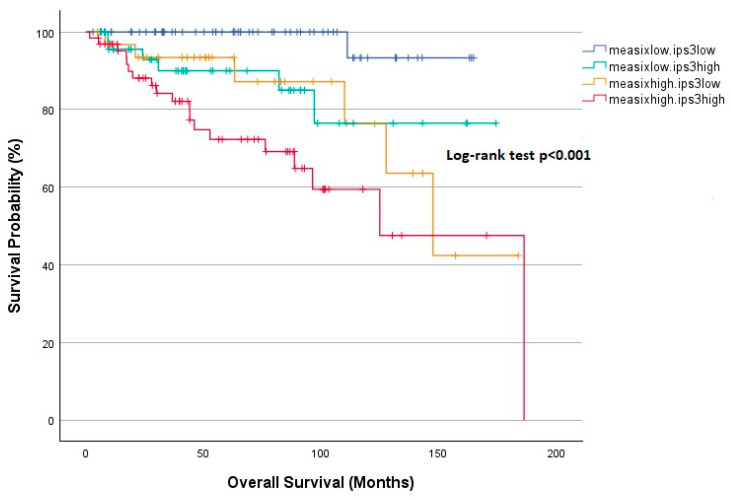
Kaplan–Meier analysis for OS according to mEASIX and IPS-3.

**Table 1 diagnostics-15-00185-t001:** General Patients Characteristics.

Parameters	*n*:	206
**Age, years (median)**	33	18–80
**Sex, male (%)**	109	52.9
**Comorbidities**		
Diabetes Mellitus, (%)	14	6.8
Hypertension, (%)	12	5.8
Coronary Artery Disease, (%)	5	2.4
Chronic Obstructive Pulmonary Disease, (%)	5	2.4
Hypothyroidism, (%)	4	1.9
**Pathological Subtype**		
Nodular Sclerosis, (%)	125	60.7
Mixed Cellularity, (%)	70	34
Lymphocyte Rich, (%)	6	2.9
Lymphocyte-depleted, (%)	5	2.4
**Presence of B Symptom, (%)**	113	54.9
**Presence of Bulky Lesion, (%)**	22	10.7
**Presence of Bone Marrow Involvement, (%)**	27	13.1
**Stage**		
Stage I, (%)	28	13.6
Stage II, (%)	71	34.5
Stage III, (%)	53	25.7
Stage IV, (%)	54	26.2
**Risk Classification of Early Stage**		
Favorable, (%)	47	47.5 ^a^
Unfavorable, (%)	52	52.5 ^a^
**Stage Group**		
Early Stage, (%)	68	33
Advanced Stage, (%)	138	67
**IPS-7**		
Low risk, (%)	129	93.5 ^b^
Intermediate risk, (%)	8	5.8 ^b^
High risk, (%)	1	0.7 ^b^
**IPS-3**		
Low risk, (%)	47	34.1 ^b^
Intermediate risk, (%)	83	60.1 ^b^
High risk, (%)	8	5.8 ^b^
**Presence of Primary Refractory Disease, (%)**	23	11.2
**First Line Therapy Combined with Radiotherapy, (%)**	46	22.3
**First Line Treatment Response**		
CR, (%)	169	82
PR, (%)	14	6.8
Progression, (%)	23	11.2
**Presence of Relaps, (%)**	27	13.1

IPS-7, international prognostic scale-7; IPS-3, international prognostic scale-3; CR, Complete Response; PR, Partial Response; ^a^, Evaluated in 99 patients; ^b^, Evaluated in 138 patients.

**Table 2 diagnostics-15-00185-t002:** Predictive power of mEASIX in differentiating survival outcome and refractory disease.

mEASIX Cut-Off: 17.28	AUC	95% Cl	Sensitivity (%)	Specificity (%)	*p*-Value
PFS	0.666	0.584–0.747	70.6	60.6	**<0.001**
OS	0.728	0.637–0.819	78.1	58.6	**<0.001**
Primary Refractory Disease	0.720	0.607–0.833	82.6	57.4	**<0.001**

mEASIX, modified endothelial activation and stress index; AUC, area under the curve; CI, Confidential Interval; PFS, progression free survival; OS, overall survival.

**Table 3 diagnostics-15-00185-t003:** Characteristics of Patients Based on mEASIX Score.

Parameters	mEASIX^low^ n:109	mEASIX^high^ n:97	*p*-Value
**Age, years (median)**	33 (18–80)	34 (18–75)	0.823 ^m^
**Gender, male (%)**	54 (49.5)	55 (56.7)	0.304 ^χ2^
**Presence of B Symptom, (%)**	33 (30.3)	80 (82.5)	**<0.001 ^χ2^**
**Presence of Bulky Lesion, (%)**	12 (11)	10 (10.3)	0.871 ^χ2^
**Presence of Bone Marrow Involvement, (%)**	8 (7.3)	19(19.6)	**0.009 ^χ2^**
**Pathological Subtype**			
Nodular Sclerosis, (%)	72 (66.1)	53 (54.6)	0.251 ^χ2^
Mixed Cellularity, (%)	31 (28.4)	39 (40.2)	
Lymphocyte Rich, (%)	4 (3.7)	2 (2.1)	
Lymphocyte-depleted, (%)	2 (1.8)	3 (3.1)	
**Stage**			
Stage I, (%)	26 (23.9)	2 (2.1)	**<0.001 ^χ2^**
Stage II, (%)	49 (45)	22 (22.6)	
Stage III, (%)	19 (17.4)	34 (35.1)	
Stage IV, (%)	15 (13.7)	39 (40.2)	
**Stage Group**			
Early Stage, (%)	60 (55)	8 (8.2)	**<0.001 ^χ2^**
Advanced Stage, (%)	49 (45)	89 (91.8)	
**IPS-7**			
Low risk, (%)	49 (100)	80 (89.9)	0.050 ^χ2^
Intermediate risk, (%)	-	8 (8.9)	
High risk, (%)	-	1 (1.2)	
**IPS-3**			
Low risk, (%)	19 (38.8)	28 (31.5)	0.308 ^χ2^
Intermediate risk, (%)	29 (59.1)	54 (60.6)	
High risk, (%)	1 (2.1)	7 (7.9)	
**Total Leukocyte, 10^9^/L (median)**	8.9 (3.5–34.5)	10.5 (1.5–38.5)	0.069 ^m^
**Neutrophil, 10^9^/L (median)**	6 (1.1–43)	7.2 (0.67–35.2)	**0.023 ^m^**
**Lymphocyte, 10^9^/L (median)**	2.1 (0.3–18)	1.5 (0.2–5.5)	**<0.001 ^m^**
**Hemoglobin, g/dl (mean ± SD)**	12.8 ± 1.91	11.1 ± 1.89	**<0.001 ^t^**
**Platelets, 10^9^/L (median)**	292 (137–619)	380 (30–972)	**<0.001 ^m^**
**Serum LDH, IU/L**	201 (119–366)	246 (108–684)	**<0.001 ^m^**
**CRP, mg/L (median)**	6 (1–49)	80 (19–341)	**<0.001 ^m^**
**ESR, mm/h (median)**	23 (1–120)	56 (8–120)	**<0.001 ^m^**
**Serum Albumin, gr/dl (median)**	4.3 (2.7–5)	3.9 (2.2–4.9)	**<0.001 ^m^**
**EASIX**	0.49 (0.16–1.56)	0.44 (0.14–12.2)	0.714 ^m^
**Presence of Primary Refractory Disease, (%)**	4 (3.7)	19 (19.6)	**<0.001 ^χ2^**
**Presence of Relaps, (%)**	10 (9.2)	17 (17.6)	**0.021 ^χ2^**
**First Line Treatment Response**			
CR, (%)	100 (91.7)	69 (71.2)	**0.001 ^χ2^**
PR, (%)	5 (4.6)	9 (9.2)	
Progression, (%)	4 (3.7)	19 (19.6)	

mEASIX, modified endothelial activation and stress index; IPS-7, international prognostic scale-7; IPS-3, international prognostic scale-3; LDH, lactate dehydrogenase; CRP, C-reactive protein; ESR, erythrocyte sedimentation rate; EASIX, endothelial activation and stress index; CR, Complete Response; PR, Partial Response; m, Mann–Whitney U test; χ^2^, Chi-squared test; t, Student’s *t*-test, Bold values; Statistically Significant.

**Table 4 diagnostics-15-00185-t004:** Univariate and Multivariate Logistic Regression Analysis for Predicting Primary Refractory Cases.

Factor	Univariate Analysis				Multivariate Analysis			
	OR	95% Cl		*p*-Value	OR	95% Cl		*p*-Value
		Lower	Upper			Lower	Upper	
**Gender (Male [RC] vs. Female)**	0.967	0.406	2.304	0.940				
**Age (years)**	0.985	0.955	1.016	0.341				
**Stage**	1.577	1.001	2.486	0.050				
**Pathological Subtype (NS vs. other)**	1.018	0.537	1.930	0.955				
**Presence of B Symptom**	4.497	1.473	13.736	**0.008**				
**Presence of Bulky Lesion**	0.776	0.169	3.559	0.744				
**Presence of Bone Marrow Involvement**	4.604	1.726	12.275	**0.002**	4.079	1.390	11.972	**0.010**
**WBC (10^9^/L)**	1.000	1.000	1.000	**0.014**	1.000	1.000	1.000	0.036
**Neutrophil (10^9^/L)**	1.000	1.000	1.000	**0.027**				
**Lymphocyte (10^9^/L)**	1.000	0.999	1.000	0.546				
**Hemoglobin (g/dl)**	0.738	0.597	0.912	**0.005**				
**Platelets (10^9^/L)**	1.000	1.000	1.000	0.086				
**Serum LDH (IU/L)**	1.006	1.002	1.009	**0.004**				
**CRP (mg/L)**	1.010	1.004	1.016	**<0.001**				
**ESR (mm/h)**	1.015	1.003	1.028	**0.017**				
**Albumin (gr/dl)**	0.910	0.846	0.980	**0.012**				
**IPS-7**	4.416	1.241	15.712	**0.022**				
**IPS-3**	1.910	0.887	4.112	0.098				
**EASIX**	1.227	0.949	1.587	0.119				
**mEASIX (High [RC] vs. low)**	6.253	2.046	19.111	**0.001**	6.563	1.826	23.593	**0.004**

OR, Odds Ratio; CI, Confidential Interval; RC, Reference Category; NS, Noduler Sclerosis; WBC, total leukocyte count; LDH, lactate dehydrogenase; CRP, C-Reactive Protein; ESR, Erythrocyte Sedimentation Rate; IPS-7, international prognostic scale-7; IPS-3, international prognostic scale-3; EASIX, endothelial activation and stress index; mEASIX, modified endothelial activation and stress index, Bold values; Statistically Significant.

**Table 5 diagnostics-15-00185-t005:** Univariate and Multivariate Cox Regression Analysis for Progression Free Survival.

Factor	Univariate Analysis				Multivariate Analysis			
	HR	95% Cl		*p*-Value	HR	95% Cl		*p*-Value
		Lower	Upper			Lower	Upper	
**Gender (Male [RC] vs. Female)**	0.988	0.570	1.712	0.965				
**Age (years)**	0.979	0.958	1.000	**0.045**				
**Stage**	1.566	1.172	2.093	**0.002**				
**Pathological Subtype (NS vs. other)**	0.637	0.387	1.049	0.076	0.484	0.278	0.845	**0.011**
**Presence of B Symptom**	2.733	1.455	5.132	**0.002**				
**Presence of Bulky Lesion**	1.649	0.774	3.512	0.195				
**Presence of Bone Marrow Involvement**	2.708	1.416	5.177	**0.003**	2.610	1.280	5.323	**0.008**
**WBC (10^9^/L)**	1.000	1.000	1.000	**0.017**				
**Neutrophil (10^9^/L)**	1.000	1.000	1.000	**0.031**				
**Lymphocyte (10^9^/L)**	1.000	1.000	1.000	0.432				
**Hemoglobin (g/dl)**	0.838	0.732	0.958	**0.010**				
**Platelets (10^9^/L)**	1.000	1.000	1.000	**0.034**	1.000	1.000	1.000	0.064
**Serum LDH (IU/L)**	1.003	1.000	1.005	**0.043**				
**CRP (mg/L)**	1.007	1.003	1.010	**<0.001**				
**ESR (mm/h)**	1.007	0.999	1.015	0.079				
**Albumin (gr/dl)**	0.956	0.912	1.003	0.066				
**IPS-7**	3.018	1.360	6.699	**0.007**				
**IPS-3**	1.406	0.873	2.263	0.161				
**EASIX**	1.048	0.864	1.271	0.633				
**mEASIX (High [RC] vs. low)**	3.366	1.841	6.154	**<0.001**	3.152	1.618	6.140	**<0.001**

HR, Hazard Ratio; CI, Confidential Interval; RC, Reference Category; NS, Noduler Sclerosis; WBC, total leukocyte count; LDH, lactate dehydrogenase; CRP, C-Reactive Protein; ESR, Erythrocyte Sedimentation Rate; IPS-7, international prognostic scale-7; IPS-3, international prognostic scale-3; EASIX, endothelial activation and stress index; mEASIX, modified endothelial activation and stress index, Bold values; Statistically Significant.

**Table 6 diagnostics-15-00185-t006:** Univariate and Multivariate Cox Regression Analysis for Overall Survival.

Factor	Univariate Analysis				Multivariate Analysis			
	HR	95% Cl		*p*-Value	HR	95% Cl		*p*-Value
		Lower	Upper			Lower	Upper	
**Gender (Male [RC] vs. Female)**	1.340	0.649	2.766	0.428				
**Age (years)**	1.015	0.992	1.038	0.192				
**Stage**	1.955	1.305	2.929	**0.001**				
**Pathological Subtype (NS vs. other)**	1.296	0.833	2.015	0.251				
**Presence of B Symptom**	4.045	1.553	10.537	**0.004**				
**Presence of Bulky Lesion**	1.774	0.680	4.629	0.241				
**Presence of Bone Marrow Involvement**	3.456	1.523	7.844	**0.003**				
**WBC (10^9^/L)**	1.000	1.000	1.000	0.197				
**Neutrophil (10^9^/L)**	1.000	1.000	1.000	0.238				
**Lymphocyte (10^9^/L)**	1.000	1.000	1.000	0.918				
**Hemoglobin (g/dl)**	0.799	0.672	0.949	**0.011**				
**Platelets (10^9^/L)**	1.000	1.000	1.000	0.711				
**Serum LDH (IU/L)**	1.004	1.001	1.007	**0.003**				
**CRP (mg/L)**	1.008	1.004	1.012	**<0.001**				
**ESR (mm/h)**	1.011	1.001	1.021	**0.024**				
**Albumin (gr/dl)**	0.928	0.877	0.983	**0.010**				
**IPS-7**	3.183	1.323	7.660	**0.010**				
**IPS-3**	2.725	1.486	4.994	**0.001**	1.945	1.038	3.643	**0.038**
**EASIX**	1.114	0.942	1.318	0.206				
**mEASIX (High [RC] vs. low)**	4.269	1.838	9.915	**<0.001**	3.890	1.546	9.786	**0.004**

HR, Hazard Ratio; CI, Confidential Interval; RC, Reference Category; NS, Nodular Sclerosis; WBC, total leukocyte count; LDH, lactate dehydrogenase; CRP, C-Reactive Protein; ESR, Erythrocyte Sedimentation Rate; IPS-7, international prognostic scale-7; IPS-3, international prognostic scale-3; EASIX, endothelial activation and stress index; mEASIX, modified endothelial activation and stress index, Bold values; Statistically Significant.

## Data Availability

The raw data supporting the conclusions of this article will be made available by the authors on request.

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
