# Peer review of "Modified Endothelial Activation and Stress Index: A New Predictor for Survival Outcomes in Classical Hodgkin Lymphoma Treated with Doxorubicin-Bleomycin-Vinblastine-Dacarbazine-Based Therapy"

_diagnostics, 2025, doi:10.3390/diagnostics15020185_

Round 1

Reviewer 1 Report

Comments and Suggestions for Authors

After analyzing the article, my comments and questions for the authors of the papers are as follows:

1. Did the patients really only have the comorbidities listed in Table 1? Statistically, at least one patient should also have some thyroid disease and hypercholesterolemia.

2. Did the patients take any immunosuppressive drugs in addition to the hematological treatment used? This should be noted in the article.

3. Did the patients have a secondary immunodeficiency, e.g. iron, folic acid and vitamin D deficiency? Was this tested?

4. Were the concentrations of the main immunoglobulin classes tested in the patients, at least IgG or a proteinogram to assess the gamma-globulin fraction, which contains antibodies?

Note: Questions 3 and 4 do not affect my opinion on the article, because the research topic is consistent with the methodology. They are dictated only by concern for patients.

A practical note for the future for the authors of the papers as clinicians:

Currently, there is a huge shortage of non-nutritional bioactive compounds in food, i.e. vitamins, micro- and macroelements, and vitamins in food due to the operation of the food industry. That is why we have a plague of people with secondary immunodeficiency even with a normal body weight. Without building blocks, the immune system will not function properly. That is why it is worth testing patients for at least vitamin D3, folic acid, vitamin B12, iron, and total iron binding capacity to determine transferrin saturation. Hematological treatment is an additional burden on an undernourished organism.

My personal note: As a clinical immunologist who sees patients with hematological diseases, it saddens me that this issue is not addressed by the Hematological Societies, because it affects treatment, the occurrence of complications, and patient survival.

Author Response

Comments-1: Did the patients really only have the comorbidities listed in Table 1? Statistically, at least one patient should also have some thyroid disease and hypercholesterolemia.

Response-1: Thank you for emphasising this point. Four patients with hypothyroidism were added to Table- 1. Twelve patients with hypercholesterolemia were included in the group with diabetes mellitus or coronary artery disease and were not listed separately in Table- 1.

Comments-2: Did the patients take any immunosuppressive drugs in addition to the hematological treatment used? This should be noted in the article.

Response-2: Thank you for this important feedback. All of the patients in the study group were not receiving any immunosuppressive treatment at the time of diagnosis because they were all primary hodgkin lymphoma cases. They did not receive any immunosuppressive treatment other than curative chemo/ chemoradiotherapy during the treatment of hodgkin lymphoma. In line with your suggestion, this situation is specified in the method section (line 135-136).

Comments-3: Did the patients have a secondary immunodeficiency, e.g. iron, folic acid and vitamin D deficiency? Was this tested?

Response-3: Thank you for your insightful feedback. We were unable to incorporate this data due to some patients in the study group lacking it, and the study design did not account for this issue. We plan to conduct a separate study to assess the significance of this aspect.

Comments-4: Were the concentrations of the main immunoglobulin classes tested in the patients, at least IgG or a proteinogram to assess the gamma-globulin fraction, which contains antibodies?

Response-4: Thank you for highlighting this important point. Unfortunately, we do not have data on the patients because quantitative examination of immunoglobulin values is not part of the routine workup for the remaining Hodgkin's lymphoma cases, as HIV-positive cases are excluded. (NCCN Guidelines for Hodgkin Lymphoma)

General Comments: A practical note for the future for the authors of the papers as clinicians: Currently, there is a huge shortage of non-nutritional bioactive compounds in food, i.e. vitamins, micro- and macroelements, and vitamins in food due to the operation of the food industry. That is why we have a plague of people with secondary immunodeficiency even with a normal body weight. Without building blocks, the immune system will not function properly. That is why it is worth testing patients for at least vitamin D3, folic acid, vitamin B12, iron, and total iron binding capacity to determine transferrin saturation. Hematological treatment is an additional burden on an undernourished organism. My personal note: As a clinical immunologist who sees patients with hematological diseases, it saddens me that this issue is not addressed by the Hematological Societies, because it affects treatment, the occurrence of complications, and patient survival.

Response: Thank you for highlighting this important point, crucial in daily treatment practice. While there are numerous studies on new treatment modalities, particularly in the era of targeted therapies, it is often overlooked that the patient's immune system plays a key role in determining prognosis. We appreciate your guidance in helping us to design future studies that will evaluate the prognostic significance of this issue based on your feedback.

Reviewer 2 Report

Comments and Suggestions for Authors

This is a concise and interesting paper describing interesting and clinically useful data, but two crucial issues should be better addressed.

A) There is no detailed explanation of the EASIX score, which was published in subspeciality journals and is probably not familiar to most readers. 

B) The predictive/prognostic potential of mEASIX should be better compared with other gold standard HL stratification tools (eg IPS-3, IPS-7, etc) in dedicated Kaplan Meier images.   

Author Response

General Comments: This is a concise and interesting paper describing interesting and clinically useful data, but two crucial issues should be better addressed.

Response: Thank you for your positive suggestions regarding the article and recommendations for improvement. Below, you will find our detailed responses to your concerns, along with the respective lines from the current version of the manuscript.

Comments-1: There is no detailed explanation of the EASIX score, which was published in subspeciality journals and is probably not familiar to most readers. 

Response-1: Thank you for your evaluation which contributed to developing the article. In the introduction section, we have rephrased EASIX score with more detail in line with your suggestion (line 60-76).

Comments-2: The predictive/prognostic potential of mEASIX should be better compared with other gold standard HL stratification tools (eg IPS-3, IPS-7, etc) in dedicated Kaplan Meier images.

Response-2: Thank you for your valuable feedback, which enhanced the article's content. The IPS-3 score is primarily used for current risk assessment in Hodgkin Lymphoma, and our study found it to be a significant predictor of overall survival in Cox regression analysis. Following your suggestion, we conducted Kaplan-Meier analyses to evaluate the combined effects of IPS-3 and mEASIX. The comparative analyses results are detailed in the results section and Figure 5 and Figure 6 included for progression-free survival and overall survival (line 276-288). Comparative data in the literature on current results are included in the disccusion section (line 343-350).

Round 2

Reviewer 2 Report

Comments and Suggestions for Authors

My comments have been addressed and the manuscript has been significantly improved.